# Characterization and Short-Term Outcome of Potential Celiac Disease in Children

**DOI:** 10.3390/medicina59071182

**Published:** 2023-06-21

**Authors:** Michal Kori, Chani Topf-Olivestone, Firas Rinawi, Raffi Lev-Tzion, Nadya Ziv-Sokolovskaya, Noy Lapidot Alon, Anat Guz-Mark, Raanan Shamir

**Affiliations:** 1Pediatric Gastroenterology, Kaplan Medical Center, Rehovot 5801303, Israel; 2Faculty of Medicine, Hebrew University of Jerusalem, Jerusalem 9112102, Israel; 3Pediatric Gastroenterology, Assuta Medical Center, Ashdod 7747629, Israel; chaniolive@gmail.com; 4Pediatric Gastroenterology, Emek Medical Center, Afula 1834111, Israel; dr.firas.rinawi@outlook.com; 5The Ruth and Bruce Rappaport Faculty of Medicine, Technion, Haifa 3200001, Israel; 6Juliet Keidan Institute of Pediatric Gastroenterology, Shaare Zedek Medical Center, Jerusalem 9103102, Israel; raffilv@szmc.org.il; 7Institute of Pathology, Kaplan Medical Center, Rehovot 5801303, Israel; nadyazi@clalit.org.il; 8Institute of Gastroenterology, Nutrition and Liver Diseases, Schneider Children’s Medical Center of Israel, Petach Tikva 4920235, Israel; noylapidot@gmail.com (N.L.A.); anatguz@gmail.com (A.G.-M.); raanan@shamirmd.com (R.S.); 9Sackler Faculty of Medicine, Tel-Aviv University, Tel-Aviv 6997801, Israel

**Keywords:** potential celiac disease, celiac disease, follow-up, children

## Abstract

*Background and Objectives*: Potential Celiac Disease (PCD) is defined by positive celiac serology without villous atrophy. We aimed to describe the short-term outcome of pediatric PCD while consuming a gluten-containing diet (GCD). *Materials and Methods*: Retrospective analysis of pediatric PCD patients continuing GCD, between December 2018–January 2022. Baseline demographics, celiac serology and duodenal biopsy results were reviewed. Follow-up data included repeated serology and biopsy results when performed. Minimum follow-up was 12 months unless celiac disease (CeD) was diagnosed earlier. *Results*: PCD was diagnosed in 90 children (71% females) with a mean age of 7.2 (range 1.8–16.5) years. Baseline anti-tissue transglutaminase (TTG) levels were above 10 times the upper limit of normal (ULN) in 17/90 (18.9%), 3–10 × ULN in 56/90 (62.2%) and 1–3 × ULN in 17/90 (18.9%). During follow-up, the mean time was 17.6 (range 5–35) months, TTG normalized in 34/90 (37.8%), was stable in 48/90 (53.3%), and increased or remained >10 × ULN in 8/90 (8.9%). In 20/90 (22.2%) patients, a repeat endoscopy was performed, leading to CeD diagnosis in 12/20 (60%). Thus, at the end of follow-up, CeD was diagnosed in 12/90 (13.3%). In patients with TTG >10 × ULN at diagnosis, TTG normalized in 5/17, decreased to 3–10 × ULN in 8/17, and remained above 10 × ULN in 4/17. *Conclusions*: During the short-term follow-up of pediatric PCD patients, less than 15% progressed to CeD. A third had normalized TTG levels, including children with TTG >10 × ULN, indicating the need for periodic serological and histological follow-up among PCD patients.

## 1. Introduction

Potential celiac disease (PCD) is defined by the presence of positive celiac serology, positive tissue transglutaminase antibodies (anti-TTG) and endomysial antibodies (EMA) with a preserved intestinal mucosa without villous atrophy (Marsh grade 0–1). It is estimated that PCD occurs in up to 10–20% of celiac-disease (CeD) patients [1,2,3]. The prevalence of PCD is higher in younger children and decreases to <5% in children older than 5 years [4]. In the latest European Society of Pediatric Gastroenterology, Hepatology and Nutrition (ESPGHAN) guidelines for the diagnosis of CeD from 2020, it has been stated that the evolvement of potential CD to CeD occurs in 33% to 100% of cases [5]. The risk of progression from PCD to CeD and the development of villous atrophy on a repeat intestinal biopsy differ between patients. Approximately a third stop producing celiac antibodies over time, indicating at least a temporal reversibility of the process [6,7]. The predictors for the development of villous atrophy are unknown, and there is controversy over the best way to follow these patients, including the interval between serology testing and obtaining intestinal biopsies.

Several studies have tried to determine the natural history of PCD and to identify risk factors for developing villous atrophy over time. Lionetti et al. [4] followed 832 newborns with a first-degree relative with CeD over 10 years. At 10 years of age, PCD was diagnosed in 26/112 (23.2%) children. A 4-year follow-up assessment after the first biopsy showed that 19/26 children had normalization of serum antibody levels. Follow-up of 23 of these children with PCD on a gluten-containing diet over 10 years demonstrated that the risk of progression to overt CeD was very low (3/23, 13%) [8]. In the largest and most recently published study by Auricchio R. et al. [7], the authors investigated the long-term outcome of 280 children with suspected celiac disease (based on anti-TTG and EMA) who continued a GCD. The cumulative incidence of progression to villous atrophy was 43% over a 12-year period. In multivariate analysis, the baseline factors most strongly associated with the development of villous atrophy were the numbers of γδ intraepithelial lymphocyte cells followed by age (older than 3 years) and homozygosity for the HLA DQB1*02. These factors identified 80% of the children who eventually developed villous atrophy.

Since the publication of the ESPGHAN guidelines for the diagnosis of CeD in 2012 [9], and mainly since the revision published in 2020 [5], an increasing proportion of children are being diagnosed with CeD using the no-biopsy approach. Theoretically, by using the no-biopsy approach, PCD might be missed, and a proportion of children may be diagnosed with CeD even though their villi are preserved. The titer of anti-TTG in PCD is usually relatively low and rarely exceeds 10 times the cut-off level. However, most studies, including the largest study performed by Auricchio R. et al. [7], did not determine the risk for developing villous atrophy in relation to the titer of celiac antibodies (anti-TTG and anti-EMA).

In our study, we aimed to (1) characterize the short-term outcome of PCD in children who remained on a GCD (2) determine the proportion of children whose TTG antibodies normalize over time as opposed to children who remain with elevated antibodies or develop CeD, as demonstrated by a repeat biopsy (3) identify risk factors for the development of CeD based on baseline data, including the levels of celiac antibodies.

## 2. Materials and Methods

We conducted a retrospective chart review of children <18 years old, diagnosed with PCD and followed on a GCD at four Pediatric Gastroenterology units in Israel between December 2018 and January 2022. The four units each diagnose and treat approximately 150 to 300 new patients with CeD per year. Baseline data at diagnosis included age, gender, anthropometrics, symptoms and signs, a family history of CeD, serology (TTG antibody and EMA antibodies levels) and biopsy results. All children were consuming gluten in their diet at the time of endoscopy, according to the statements of the patients and parents. Not all patients were tested with the same serology assay: The Multiplex TTG (Bioplex 2200) assay was used for the majority patients [10], an ELISA assay (Thermo Fisher, Phadia, Uppsala, Sweden) for the minority. Values were described in relation to the upper limit of normal (times the upper limit of normal, × ULN), as previously reported [11]. Not all samples were tested with the same EMA assay at the same laboratory, and thus results are reported as positive or negative without the titer or degree of EMA elevation. Serology testing was performed at a maximum of 3 months prior to endoscopy. Patients with IgA deficiency were excluded. In all patients, at least six duodenal biopsies were taken, including at least one biopsy from the duodenal bulb. A pathologist with expertise in gastrointestinal pathology examined all the duodenal biopsies in an un-blinded manner. Biopsies with a Marsh 0 or 1 classification were reviewed un-blinded by the pediatric gastroenterologist and pathologist to confirm diagnosis of PCD. The World Health Organization anthropometric calculator was used to obtain the height-for-age Z-score (HAZ) and weight-for-age Z-score (WAZ). HLA genotyping was not available for this study. Follow-up data were collected while patients were on a GCD. Patients were followed at 6-month intervals for a minimum of 12 months, unless celiac disease (CD) was diagnosed earlier.

Serological, endoscopic and histologic findings during follow-up, while consuming a GCD, were collected and analyzed. The indications for repeat endoscopy included the rise in TTG to above ×10 ULN, clinical symptoms requiring a repeat endoscopic evaluation, or TTG remaining above ×10 ULN after two years.

The study was performed in line with the Declaration of Helsinki and was approved by the Regional Ethics Committee of each participating medical center.

### Statistics

Categorical variables were reported as counts and percentages, and continuous variables were reported as means ± standard deviations. Continuous variables in the various study groups were tested for normality by Shapiro–Wilk test. For variables with abnormal distribution, the Mann–Whitney non-parametric test was performed to compare between groups. Categorical variables were analyzed by Pearson’s chi-square (χ^2^) test. *p* value < 0.05 was considered statistically significant. A logistic regression model containing demographic data (age, gender) and laboratory data (TTG, hemoglobin and ferritin levels) was generated. A logistic regression model was built to identify variables predictive of the development of CeD. Data were analyzed using SPSS25.

## 3. Results

During the study period, 90 children were diagnosed with PCD; 64 (71%) were females, with a mean age of 7.2 years (range 1.8–16.5 years). At the time of endoscopy, 24 patients (27%) were asymptomatic; most of these patients were tested due to a family history of CeD. Gastrointestinal symptoms were present in 37 (41.6%), growth delay (failure to gain weight/height, short stature, delayed puberty) in 16 (18%) and iron-deficiency anemia in 7 (7.9%). A family history of CeD was present in 34 (37.8%) patients, 18 (20.2%) had a first-degree relative with CeD, and 16 (18%) had a second-degree relative with CeD. The mean hemoglobin level was 12.2 mg% (6.8–15.8 mg%), and the mean ferritin blood level was 26.9 ng/mL (3–77 ng/mL) (Table 1).

At the time of the primary endoscopy, TTG levels were above ×10 ULN in 17/90 (18.9%), between ×3–10 ULN in 56/90 (62.2%) and between ×1–3 ULN in 17/90 (18.9%) patients. EMA was positive in 50 (56.2%) patients. Duodenal biopsies demonstrated Marsh 0 in 45 (50%) and Marsh 1 in 45 (50%) patients.

All children remained on a gluten-containing diet during follow-up and had at least one TTG test performed and a GI clinic visit. They were followed for at least 12 months unless CeD was diagnosed earlier. The mean follow-up time was 17.6 months (range 5–35 months). TTG levels normalized in 34/90 (37.8%), declined or remained unchanged in 48/90 (53.3%), and increased or remained > ×10 ULN in 8/90 (8.9%) patients. (Figure 1).

A repeat endoscopy was performed in 20 patients, including four patients in which the endoscopy was done before 12 months of follow-up. The decision to perform a repeat endoscopy was based on the rise in TTG to above ×10 ULN, clinical symptoms requiring a repeat endoscopic evaluation, or TTG remaining above ×10 ULN after two years of follow-up. In 12/20 (60%) patients, the repeat biopsies revealed a Marsh 3 lesion, and CeD was diagnosed. EMA was positive in only six of these children. Thus, at end of follow-up, 12/90 children (13.3%) were diagnosed with CeD.

In the subgroup of 17 patients with TTG > ×10 ULN at the first endoscopy, 15/17 (88%) were EMA-positive. These 15 children could have been diagnosed based on the no-biopsy approach with CeD. However, most were asymptomatic and underwent the endoscopy due to the parents’ request after offering the no-biopsy versus the biopsy approach. In these 17 children, TTG normalized in 5 (29.4%), declined but remained between ×3–10 ULN in 8 and remained above ×10 ULN in 4. Five of the eight children in which TTG remained > ×10 ULN had a repeat endoscopy, and CeD was diagnosed in three of them.

The logistic regression model identified older age and TTG levels above X10 ULN as positive predictors for the development of CeD. In patients with a TTG level above ×10 ULN compared to TTG between ×1–3 ULN, the OR for developing CeD was significantly increased (OR = 20.62 (95%CI 1.05–403.5, *p* = 0.046). The risk of developing CeD increased with age by 1.22 for every additional year (OR = 1.22 (95%CI 1.01–1.49 *p* = 0.043) (Table 2). Anthropometry, sex, hemoglobin and ferritin levels did not contribute to the prediction.

## 4. Discussion

Our study demonstrated that over a short follow-up period, a mean of 18 months, PCD developed to CeD in only 12 children (13.3%), based on repeat TTG levels and repeat biopsies performed in 20 patients. In 34 (37.8%) patients, TTG normalized, and thus there was no indication for a repeat biopsy. However, clinical follow-up and repeat serology testing were recommended for all these children. In 48 (53.3%) children, TTG levels were stable. In most of these patients, there was no indication for performing a repeat biopsy during the short-term follow-up period, which was less than 2 years. All these children require continued close follow-up and a repeat endoscopy with biopsy if their TTG levels do not normalize or there are clinical indications for re-evaluation. While high TTG titers as well as older age were identified as positive predictors for the development of CeD, TTG normalized in 38% of patients, including in 29% of children who had TTG levels above X10 UNL upon diagnosis. Even among the 20 patients who had a repeat endoscopy performed during follow-up, CeD was diagnosed in only 12 (60%) of them.

Definitive predictors for the development of CeD from PCD are lacking. The amount and the type of intraepithelial lymphocytosis (γδ) and the presence of certain HLA-DR and DQ genotyping [6] have been shown to serve as good markers for progression to CD; however, they are not available in most centers. Other studies suggest the presence of EMA IgA antibodies as a predictor [12,13]. In our study, the presence of IELs (Marsh 1 histological finding) was not associated with progression to CeD, neither was EMA positivity. In a recent small study by Sakhuja S. et al., which followed 11 children with PCD who progressed to CeD, the authors demonstrated lower anti-TTG IgA concentrations (2.4 (1.6–5) × ULN) vs. 6.41 (3.4–10.5) × ULN) at the time of the first biopsy in PCD patients compared to patients diagnosed with CeD [14]. However, probably due to the small sample size, the TTG level could not predict the development of CeD among their cohort.

Our logistic regression model identified two predictors for the development of CD: TTG levels above ×10 ULN compared to TTG × 1–3 ULN, and older age (the older the child was at the time of the first endoscopy, the higher the risk of developing CeD). While age and TTG level may be helpful in determining the interval between serology testing and intestinal biopsies, it is worth noting that during the short period of follow-up, TTG levels normalized in 34 (37.8%), declined or remained unchanged in 48 (53.3%), and increased or remained > ×10 ULN in 8 (8.9%) patients. This suggests that upon identifying elevated TTG levels, an expectant approach is warranted, especially in asymptomatic cases [2,15].

Since the publication of the ESPGHAN guidelines for the diagnosis of CD in 2012 [9], and especially since the revision published in 2020 [5], many children are being diagnosed with CeD based on serology testing without endoscopy and biopsy. The latest guidelines enable diagnosing CeD when TTG levels are above ×10 ULN with a positive EMA in a second sample, in symptomatic and asymptomatic patients. Using the current guidelines, PCD may be misdiagnosed as CeD, committing children to a lifelong GFD even though some may not have intestinal mucosal damage. The risk of diagnosing PCD decreases as the level of TTG increases; however, there are patients with TTG levels above ×10 ULN without villous atrophy. In our study, 17 PCD patients had an initial TTG level above ×10 ULN (EMA positive in 88%); thus, some of these patients might have been diagnosed with CeD if the biopsy would have been omitted. In two cases, a repeat biopsy did not demonstrate villous atrophy even though TTG levels remained high. As the risks of micronutrient deficiency, growth impairment and the development of autoimmune disease in PCD continuing a GCD are unknown, longer follow-up is essential to answer these questions. The role of active interventions, to modulate the immune response to gluten or to influence mucosal permeability with the use of probiotics [16], should also be explored in patients with PCD.

The strengths of our study include the relatively large number of children with PCD diagnosed and followed at four Pediatric Gastroenterology units in Israel. All children had at least one follow-up visit including TTG levels. All patients were followed for at least 12 months unless CeD was diagnosed earlier. The limitations of the study include its retrospective design and the relative short-term follow-up of 1–3 years. The fact that routine repeated endoscopies were not performed in all patients and only a small number of patients had repeat biopsies may be considered a major limitation; however, in 37.8%, TTG normalized, and thus there was no indication for a repeat endoscopy. In 53.3% of children, TTG levels were stable, and a repeat biopsy is planned during continued follow-up, when indicated. We also were unable to determine the percentage of PCD patients out of the total number of CeD patients diagnosed during the study period, since the number of patients undergoing endoscopy for suspected CeD during the study period was not available at all centers. However, all the centers included in the study diagnose a high number of new CeD patients per year, ranging from 150 to over 300 patients per center. Lastly, centers were using different serological kits with different performances. However, the results are in accordance with what is expected in real life, and as all kits used in Israel have a sensitivity and specificity above 90%, we suspect that using one serological kit would not result in a significant change in the trends observed in this study.

## 5. Conclusions

In conclusion, children with PCD may be followed on a GCD, since most children will not develop CeD in the short-term follow-up. Over a third of children will normalize their TTG levels within a year. PCD may be diagnosed even in children with TTG levels above ×10 ULN, although the long-term outcome of this subgroup should be further investigated. Children with PCD should undergo close monitoring of clinical and laboratory parameters, including repeated endoscopy when appropriate.

## Figures and Tables

**Figure 1 medicina-59-01182-f001:**
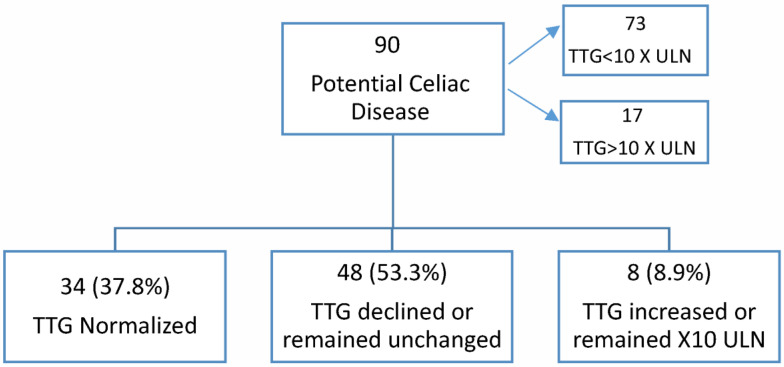
Short-term follow-up of TTG levels of PCD patients. TTG—tissue transglutaminase, ULN—Upper Limit of Normal.

**Table 1 medicina-59-01182-t001:** Characteristics and baseline data of 90 Potential Celiac Disease patients.

Age (Years) (Mean, Range)		7.2 (1.7–16.5)
Sex (Female)		64 (71.1%)
Symptoms leading to diagnosis	Gastrointestinal	37 (41.1%)
	Growth delay	16 (17.8%)
	Iron-deficiency anemia	7 (7.8%)
	No symptoms	25 (27.8%)
Family history of CeD	Positive	34 (37.8%)
	1st degree	18 (20%)
	2nd degree	16 (17.8%)
Hemoglobin level (gr%) (mean, range)		12.2 (6.8–15.6)
Ferritin (ng/mL) (mean, range)		26.9 (3–77)

**Table 2 medicina-59-01182-t002:** Logistic regression analysis to identify predictors of Celiac Disease.

	OR	95% CI		*p* Value
Age (years)	1.22	1.01	1.49	0.043
TTG × 1–3 ULN				0.127
TTG × 3–10 ULN	4.35	0.39	48.82	0.233
TTG above ×10 ULN	20.62	1.05	403.51	0.046
Hemoglobin	1.34	0.07	2.69	0.408
Ferritin	1.02	0.99	1.06	0.233
Gender (female)	0.68	0.10	4.50	0.693

TTG—Tissue Transglutaminase, ULN—Upper Limit of Normal, CI—confidence interval, OR—odd ratio; *p*-values obtained from the Wald Chi-Square Test for the significance of OR.

## Data Availability

Data is available upon reasonable request.

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
