# Peer review of "Characterization and Short-Term Outcome of Potential Celiac Disease in Children"

_medicina, 2023, doi:10.3390/medicina59071182_

Round 1

Reviewer 1 Report

In this study the authors have analysed the course of potential coeliac disease, which is a topic of interest for many within the field.

I have read it with great interest, but would suggest a couple of things in order to improve both the quality of the scientific content and its readability.  

Point by point revision:

 Introduction

The introduction contains quite some repetitive elements and is quite long. For example, the paper of Auricchio is mentioned 2 times more or less with the same content on page 2 lines 47-50 and from lines 58-67.

Furthermore, for readers not so familiar with coeliac disease, it could be more apprehensive if the authors mention the proportion of possible potential CD due to the ESPGHAN 2020 guidelines earlier.

 Methods

The methods could be a bit more clear: how did the authors know that the children at time of biopsy stil late gluten? This is important since this could have explained why histology was normal.

Furthermore, the time between serology and biopsy isn’t described as is the fact when review of Marsh 0-1 was done by a ped GI and pathologist together. Was it done retrospectively for the purpose of this study or at the time of diagnosis? And who were these professionals? Were they blinded for the results or not?

Was there a follow-up protocol during the study period for potential CD, in other words: how was decided who got a FU biopsy? This is unclear. Do we know whether they went on a GFD during follow-up? This is especially relevant for the group who’s Abs got lower or normal.  

 Results

How many CD children were there during the study period? This is relevant to know how experienced the study team is.

And why were the children with 10xULN biopsied in the first place? Was it done because they were asymptomatic or perhaps type 1 DM patients?

Perhaps a figure with all the numbers per titer, percentages of antibody levels over time and final diagnosis could be considered, because now it is pretty hard to really understand the numbers.

 Discussion

The first part of the discussion is in part a repitition of the introduction, it can be shortened.

The percentage of patients with TGA> 10xULN in the potential CD group seems high, but needs to be interpreted with care since, as the authors state, the total number of CD patients in the entire CD-group is unknown. With a known sens/spec of 95-96% of TG2A there should be at least 340 children in all centers. If less, there could be doubt about the testkit or the histology.

Author Response

Reviewer 1

We would like to thank the reviewers for their timely comments and suggestions. A point by point reply to each of the comments is outlined below with a revised manuscripts with all changed tracked. 

In this study the authors have analysed the course of potential coeliac disease, which is a topic of interest for many within the field.

I have read it with great interest, but would suggest a couple of things in order to improve both the quality of the scientific content and its readability.  

Point by point revision:

 Introduction

The introduction contains quite some repetitive elements and is quite long. For example, the paper of Auricchio is mentioned 2 times more or less with the same content on page 2 lines 47-50 and from lines 58-67.

Reply – The introduction was shortened as suggested, and repetitive elements were omitted.

Furthermore, for readers not so familiar with coeliac disease, it could be more apprehensive if the authors mention the proportion of possible potential CD due to the ESPGHAN 2020 guidelines earlier.

Reply – We added the proportion of possible potential CD as stated in the latest ESPGHAN 2020 guidelines in the beginning of the introduction.

 Methods

The methods could be a bit more clear: how did the authors know that the children at time of biopsy still ate gluten? This is important since this could have explained why histology was normal.

Reply- We questioned parents and children about the consumption of gluten at the time of endoscopy. This was added to the methods section. All children were consuming gluten in their diet, at the time of endoscopy, as reported by the patients and parents. Children on a GFD were excluded.

Furthermore, the time between serology and biopsy isn’t described as is the fact when review of Marsh 0-1 was done by a ped GI and pathologist together. Was it done retrospectively for the purpose of this study or at the time of diagnosis? And who were these professionals? Were they blinded for the results or not?

Reply- The time between serology testing and biopsy was less than 3 months. This was added to the method section. Serology testing was performed at a maximum of 3 months prior to endoscopy.

The Pediatric GI specialist and pathologist together, as a common practice in all cases of PCD (not for the study, which was retrospective) did the review of all biopsies with Marsh 0-1 un-blinded. All professionals are experienced pathologists and Ped GI doctors. We added that the review of biopsies was un-blinded.

Was there a follow-up protocol during the study period for potential CD, in other words: how was decided who got a FU biopsy? This is unclear. Do we know whether they went on a GFD during follow-up? This is especially relevant for the group who’s Abs got lower or normal. 

Reply- This was a retrospective study. All centers participating in the study followed patients at 6-month intervals as common practice. The indication for repeat endoscopy included; a rise in TTG > X10 ULN, clinical symptoms requiring a repeat endoscopic evaluation and /or TTG remaining above X10 ULN after a two years follow-up period. This was added to the methods. All patients were on a GCD during follow-up.

 Results

How many CD children were there during the study period? This is relevant to know how experienced the study team is.

Reply- Each of the four participating centers is experienced with celiac disease. The number of new CeD patients per center ranges between 150-300 children per year.

And why were the children with 10xULN biopsied in the first place? Was it done because they were asymptomatic or perhaps type 1 DM patients?

Reply- In the clinical practice of all centers the Pediatric gastroenterologists discusses the options of biopsy-based versus no-biopsy diagnosis of CeD with patients and families, according to ESPGHAN guidelines that permit the two approaches in patients with high titers. Most of the patients with TTG 10xULN that underwent endoscopy and biopsy were asymptomatic, there were no cases type 1 DM. The reason for endoscopy was the parents' request, favoring the biopsy approach.

Perhaps a figure with all the numbers per titer, percentages of antibody levels over time and final diagnosis could be considered, because now it is pretty hard to really understand the numbers.

Reply- we had prepared a figure, which was omitted from the manuscript, figure 1. I hope the figure is sufficient

Figure 1 – Short term follow-up of TTG levels of PCD patients

There is a problem copying the figure to the reply. It will be send again as a document

 Discussion

The first part of the discussion is in part a repitition of the introduction, it can be shortened.

Reply-The first part is the results of our study. We omitted repetition of statements that were sated in the introduction thus the discussion was shortened.

The percentage of patients with TGA> 10xULN in the potential CD group seems high, but needs to be interpreted with care since, as the authors state, the total number of CD patients in the entire CD-group is unknown. With a known sens/spec of 95-96% of TG2A there should be at least 340 children in all centers. If less, there could be doubt about the testkit or the histology.

Reply- As mentioned above the four participating centers are experienced with celiac disease. The number of new CeD patients per center: is approximately 150-300 new cases in each center per year.

Reviewer 2 Report

Dear Authors,

 this is a very interesting work. I have minor suggestions:

-line 42: “Older children”?Maybe it is better to write teenagers?Or children aged >7 years etc.?

- line 69 extra space

-142: based on; wrong punctuation

To date, there is no therapy for CD and GFD is the only option. However, patients with CD or with the risk to develop CD may benefit from probiotics. Pivotal studies on the supplementation of GFD with probiotics, such as Bifidobacterium and Lactobacilli, reported a potential to restore gut microbiota composition and to pre-digest gluten in the intestinal lumen, reducing the inflammation associated with gluten intake, the intestinal permeability, and the cytokine and antibody production. These findings could explain an improvement in symptoms and quality of life in patients treated with GFD and probiotics. So, probiotics supplementation could be a strategy to keep health the mucosal barrier also in children with PCD. (Marasco G, Cirota GG, Rossini B, Lungaro L, Di Biase AR, Colecchia A, Volta U, De Giorgio R, Festi D, Caio G. Probiotics, Prebiotics and Other Dietary Supplements for Gut Microbiota Modulation in Celiac Disease Patients. Nutrients. 2020 Sep 2;12(9):2674. doi: 10.3390/nu12092674. PMID: 32887325; PMCID: PMC7551848.)

Best Regards.

Good.

Author Response

We would like to thank the reviewers for their timely comments and suggestions. A point by point reply to each of the comments is outlined below with a revised manuscripts with all changed tracked. 

Dear Authors,

 This is a very interesting work. I have minor suggestions:

-line 42: “Older children”? Maybe it is better to write teenagers ?Or children aged >7 years etc.?

Reply-

This was corrected

- line 69 extra space

Reply- This was corrected

-142: based on; wrong punctuation

Reply- This was corrected

To date, there is no therapy for CD and GFD is the only option. However, patients with CD or with the risk to develop CD may benefit from probiotics. Pivotal studies on the supplementation of GFD with probiotics, such as Bifidobacterium and Lactobacilli, reported a potential to restore gut microbiota composition and to pre-digest gluten in the intestinal lumen, reducing the inflammation associated with gluten intake, the intestinal permeability, and the cytokine and antibody production. These findings could explain an improvement in symptoms and quality of life in patients treated with GFD and probiotics. So, probiotics supplementation could be a strategy to keep health the mucosal barrier also in children with PCD. (Marasco G, Cirota GG, Rossini B, Lungaro L, Di Biase AR, Colecchia A, Volta U, De Giorgio R, Festi D, Caio G. Probiotics, Prebiotics and Other Dietary Supplements for Gut Microbiota Modulation in Celiac Disease Patients. Nutrients. 2020 Sep 2;12(9):2674. doi: 10.3390/nu12092674. PMID: 32887325; PMCID: PMC7551848.)

Reply- We added a comment concerning probiotics including the above reference. “The role of active interventions to modulate the immune response to gluten or to influence mucosal permeability with the use of probiotics (16), should further be explored in patients with PCD.”

Round 2

Reviewer 1 Report

rebuttal done succesfully, I am satisfied.